# QueST: Incentivizing LLMs to Generate Difficult Problems

## Abstract

Large Language Models have achieved strong performance on reasoning tasks, solving competition-level coding and math problems. However, their scalability is limited by human-labeled datasets and the lack of large-scale, challenging coding problem training data. Existing competitive coding datasets contain only thousands to tens of thousands of problems. Previous synthetic data generation methods rely on either augmenting existing instruction datasets or selecting challenging problems from human-labeled data. In this paper, we propose QueST, a novel framework which combines difficulty-aware graph sampling for prompt and difficulty-aware rejection fine-tuning that directly optimizes specialized generators to create challenging coding problems. Our trained generators demonstrate superior capability at creating challenging problems compared to even proprietary models such as `GPT-4o`. We leverage this method to generate large-scale synthetic coding problems, which we then use to distill from long Chain-of-Thought (CoT) models or conduct reinforcement learning for smaller models, proving effective in both scenarios. Our distilled model achieves the best performance compared to similarly sized models trained on previous long CoT SFT datasets. By training generators to create more difficult problems, QueST pushes the boundaries of reasoning abilities in large language models.

## 1 Introduction

Test-time scaling through long chain-of-thought and large-scale reinforcement learning has dramatically boosted the reasoning ability of large language models, enabling LLMs to solve competition-level coding and math problems that were previously beyond their reach. Models like OpenAI o1 (OpenAI, 2024) and DeepSeek-R1 (Guo et al., 2025) have demonstrated remarkable performance on challenging benchmarks such as Codeforces, AIME, and IOI, achieving expert-level problem-solving capabilities through extensive reasoning traces that can span thousands of tokens. However, the problems used for training these models even require expert-level human annotations, which severely limits the scalability of LLM training. Current competitive coding datasets (Li et al., 2022; 2023) contain only thousands to tens of thousands of problems. As LLMs become more capable, the requirements for training data grow increasingly demanding—often requiring PhD-level experts in mathematics, computer science, and algorithm design to propose novel problems that can genuinely challenge these models. This process is not only extremely costly for requiring experts to propose difficult problems but also fundamentally cannot scale in terms of both dataset size and problem difficulty, creating a critical bottleneck in the development of next-generation reasoning models.

To mitigate this problem, methods for synthetic data generation and augmentation have been proposed. Previous works have focused either on paraphrasing-based augmentation (Luo et al., 2025a; Yu et al., 2024) or extracting concepts and recombining them based on co-occurrence probabilities (Tang et al., 2024; Zhao et al., 2025). Some recent works have proposed leveraging model weaknesses and extracting concepts to create new problems (Liang et al., 2025). More recently, reasoning-based LLMs have presented the next paradigm in advancing large language model reasoning capabilities (OpenAI, 2024; Guo et al., 2025). Works like Guo et al. (2025); Guha et al. (2025) created long chain-of-thought responses from reasoning models and curated synthetic SFT datasets, effectively helping small open-weight LLMs achieve superior performance in code and math tasks. Ahmad et al. (2025) curated the largest open-source dataset by obtaining long CoT responses from DeepSeek-R1 multiple times for each problems, though the problems themselves are still sourced from human-

Table 1: Comparison between representative code reasoning datasets.

| Code Datasets | #Problems | Long CoT Responses | Synthetic Problems |
|---|---|---|---|
| CodeContest (Li et al., 2022) | 13K | ✗ | ✗ |
| TACO (Li et al., 2023) | 26K | ✗ | ✗ |
| Bespoke-Stratos (Labs, 2025) | 17K | ✓ | ✗ |
| Open-R1 Codeforces-cots (Face, 2025) | 10K | ✓ | ✗ |
| OpenCodeReasoning (Ahmad et al., 2025) | 28K | ✓ | ✗ |
| Ours (QueST) | **100K** | ✓ | ✓ |

labeled competition coding problems. These methods have narrowed the performance gap between open-weight models and closed-weight reasoning models.

However, existing methods described above either focus on leveraging existing human-annotated problems and curating synthetic responses from existing reasoning models, or rely on a *fixed* LLM to generate new problems by prompting. In this paper, we are the first to propose a method that directly trains an LLM generator to create challenging competitive code reasoning problems. We call our method **QueST**, embarking on a quest to generate increasingly challenging code problems through the combination of difficulty-aware graph sampling and difficulty-aware rejection fine-tuning. This approach is more scalable and flexible compared to previous methods that used a fixed generator or fixed human-labeled problems. Our proposed method makes the generator specialized and stronger than even closed-weight strong instruction models at creating challenging problems. We leverage this to generate the largest-scale code problem training set compared to previous synthetic data approaches, and the statistics of our synthetic data with previous data are shown in Table 1. We obtained responses from long chain-of-thought reasoning models, then leveraged the generated datasets to SFT small models, achieving competitive scores for similar-sized models on code reasoning benchmarks like LiveCodeBench (Jain et al., 2025) and USACO (Shi et al., 2024).

Our contributions can be summarized as follows:

- We introduce a novel difficulty-aware coding problems generation framework that combines both difficulty-aware graph sampling and difficulty-aware rejection fine-tuning, which trains specialized generators to create challenging coding problems.

- We create the largest synthetic code reasoning problem set to date, comprising over 100K challenging coding problems paired with detailed chain-of-thought solutions from reasoning models.

- We demonstrate that small models fine-tuned on our synthetic dataset achieve competitive performance among similarly-sized models on LiveCodeBench and USACO benchmarks. We also demonstrate effectiveness in RL experiments.

- We conduct comprehensive ablation studies and analyses of our proposed method and the distribution of the generated coding problems.

## 2 QUEST

In this section, we present QUEST, our proposed method for generating difficult problems. We focus our investigation on the generation of coding problems, as other forms of reasoning tasks (e.g., mathematical reasoning) can be regarded as special cases of coding tasks (Jiao et al., 2025). We begin by introducing our scaffolding framework for problem generation, which builds upon MathScale (Tang et al., 2024). Next, we detail our strategies for incentivizing LLMs to produce more difficult problems. Finally, we demonstrate how our scaffolding can be adapted to further enhance the generation of challenging problems.

### 2.1 PRELIMINARY: PROBLEM GENERATION THROUGH CONCEPT GRAPH

Our scaffolding for problem generation is based on (Tang et al., 2024), which generates new problems based on existing seed problems by prompting an LLM in three steps (i.e., concept extraction, graph construction and problem generation).

**Concept Extraction**  For each problem $q$ in the seed problem set $\mathbf{Q}_{\text{seed}}$, we prompt an LLM to extract concepts $c$ (topics and knowledge points) from it. We follow the setting of Tang et al. (2024), topics refers to general directions, knowledge points refers to more fine-grained concepts, example can be found in Appendix Table 5. Note that problem generation can be guided later using the concepts that we extracted in this step. The process can be defined formally as follows:

$$\mathbf{C} = \mathcal{G}^{\text{Q}}((p_{\text{extract}}, \mathbf{Q}_{\text{seed}})) \tag{1}$$

where $p_{\text{extract}}$ is the prompt used to extract concepts, $\mathcal{G}^{\text{Q}}$ is our problem generator and $\mathbf{C}$ is the set of concepts extracted. Detail prompts of $p_{\text{extract}}$ are in Appendix Table 7.

**Graph Construction**  Once we obtain the concepts $\mathbf{C}$, we proceed to identify plausible combinations of these concepts. Two concepts are considered to form a reasonable combination if they have frequently co-occurred within the same problem in the seed dataset. To capture these relationships, we construct a concept graph in which nodes represent individual concepts, and edge weights encode the strength of co-occurrence between concept pairs. The edge weight $w(u, v)$ is defined as follows:

$$w(u, v) = \log\left(\text{freq}(u, v) + \epsilon\right) \tag{2}$$

where $u$ and $v$ denote concept nodes, and $\text{freq}(u, v)$ represents the observed co-occurrence frequency of these concepts. A small constant $\epsilon$ is added to ensure numerical stability by preventing zero counts.

Given the constructed graph, we proceed to sample concept combinations, which are then utilized for the generation of new problems. We start from a uniformly random sampling from all the topics and subsequently perform up to six steps of a random walk on the graph (Tang et al., 2024). At each step, the transition probability from node $\mathbf{u}$ to node $\mathbf{v}$ is defined as:

$$p_{\mathbf{u},\mathbf{v}} = \frac{\exp\left(w(\mathbf{u}, \mathbf{v})\right)}{\sum_{\mathbf{v}' \in \mathcal{N}(\mathbf{u})} \exp\left(w(\mathbf{u}, \mathbf{v}')\right)} \tag{3}$$

where $\mathcal{N}(\mathbf{u})$ denotes the set of nodes adjacent to u. After each random walk episode, we obtain a sampled concept combination $s$, which is subsequently used for problem generation.

**Problem Generation**  Given the sampled concept set $s$, we leverage an LLM to generate new problems. We incorporate few-shot examples to guide the LLM in formulating problems. These examples are selected from the pool of seed problems based on the Jaccard distance between their respective sets of concepts. Formally, this process can be described as:

$$\mathbf{Q}_{\text{new}} = \mathcal{G}^{\text{Q}}(p_{\text{generate}}, \mathcal{S}(\mathbf{C}), \mathbf{Q}_{\text{seed}}) \tag{4}$$

where $\mathcal{S}(\mathbf{C})$ denotes the set of sampled concepts, and $p_{\text{generate}}$ represents the prompt template utilized for problem generation (see Appendix Table 5 for additional details).

At this stage, our problem generator is designed to produce new problems, rather than explicitly targeting increased difficulty. The generation of more challenging problems will be addressed in the subsequent sections.

## 2.2 DIFFICULTY-AWARE REJECTION FINETUNING

We focus primarily on the generation of challenging coding problems, though our approach is readily extensible to other forms of reasoning tasks. We first present our method for measuring problem difficulty, and then illustrate how this measure is employed to guide LLMs in producing more difficult problems.

**Difficulty Estimation**  A natural way to assess the difficulty of a generated problem is to examine the consistency of the models multiple outputs. Wang et al. (2023a) finds that self-consistency is highly correlated with accuracy, which reflects the uncertainty of the models, and also the difficulty of the problem. When most solutions converge to a single outcome, the problem is likely straightforward. Conversely, if the solutions diverge and produce inconsistent outputs, this indicates model uncertainty, suggesting the problem is more difficult.

Building on this intuition, we define the difficulty of a problem using the *average majority voting rate* of its solutions. We illustrate our metric using coding problems as a case study, noting that

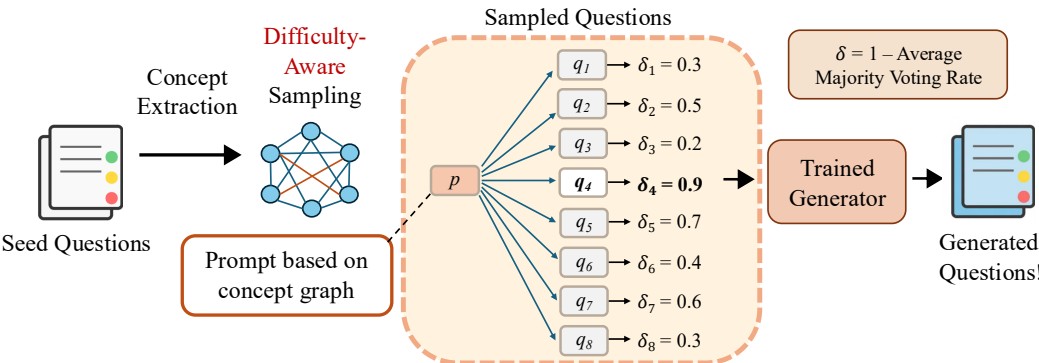

Figure 1: The pipeline of QueST. We first extract concepts based on seed problems, then use difficulty-aware sampling method described in Equation 10 to create prompts for problem generation. We generate 8 problems for each prompt, calculate the difficulty $\delta$ of the generated problem based on Equation 6, and use the most difficult problem as rejection fine-tuning data to train our generator.

other verifiable reasoning problems (e.g. math) can be regarded as a special case of this setting. Let $q \sim \mathcal{G}^Q(p_{\text{generate}}, s, \mathbf{Q}_{\text{seed}})$ denote a generated coding problem (see Equation (4)), where $s \sim \mathcal{S}(\mathbf{C})$ is a sampled concept combination (see Section 2.1). The estimation proceeds in three steps. First, we prompt gpt-4o to generate $T$ test inputs, forming the set $\mathcal{I} = \{i_1, i_2, \ldots, i_T\}$ (details of the prompt in Appendix Table 6). Second, we obtain $M$ candidate solutions $\mathcal{Y} = \{y_1, y_2, \ldots, y_M\}$ from gpt-4o. Third, we execute each $y_m \in \mathcal{Y}$ on all inputs $i_t \in \mathcal{I}$, producing output sets $\mathcal{O}_t = \{g(y_1, i_t), g(y_2, i_t), \ldots, g(y_M, i_t)\}$, where $g(y_m, i_t)$ denotes extracting the code from $y_m$, running it on input $i_t$, and recording the output. For each test input, the most likely output $o_t$ is identified as the most frequent element in $\mathcal{O}_t$:

$$o_t = \arg\max_{o \in \mathcal{O}_t} f(o, \mathcal{O}_t) \tag{5}$$

where $f(o, \mathcal{O}_t) = \left|\{x \in \mathcal{O}_t \mid x = o\}\right|$ counts the occurrences of $o$ in $\mathcal{O}_t$. Finally, we quantify the problem difficulty as

$$\delta(q) = 1 - \frac{1}{T}\sum_{t=1}^{T} \frac{f(o_t, \mathcal{O}_t)}{M} \tag{6}$$

Intuitively, $\delta(q)$ measures the degree of disagreement among candidate solutions: the lower the majority voting rate, the higher the difficulty. Thus, larger values of $\delta(q)$ correspond to more challenging problems.

To further enhance the probability of generating valid synthetic problems, we filter out problems where over half of the test case outputs from generated responses return None, indicating unsuccessful code execution.

**Rejection Fine-tuning**  Having introduced the difficulty measure $\delta(q)$, we now describe how it is employed to construct a dataset of prompt–problem pairs for training LLMs to generate difficult problems. The key idea is to sample multiple candidate problems from the same prompt and retain only the most difficult one.

As discussed in Section 2.1, for each concept combination $s$, a problem can be generated via

$$q \sim \mathcal{G}^Q(p_{\text{generate}}, s, \mathbf{Q}_{\text{seed}}) \tag{7}$$

where $\mathcal{G}^Q$ denotes the LLM-based generator. More generally, let $M_\theta$ be the LLM parameterized by $\theta$, and let $p$ denote the *actual* prompt ($p_{\text{generate}}$ instantiated with concept set $s$ and seed problems $\mathbf{Q}_{\text{seed}}$) used to query $M_\theta$. By sampling $K$ times, we obtain a set of candidate problems:

$$q_k \sim M_\theta(p) \quad \text{for } k = 1, \ldots, K \tag{8}$$

We denote this set by $\mathcal{Q} = \{q_1, q_2, \ldots, q_K\}$. We then select the most difficult problem according to our measure $\delta(\cdot)$:

$$q^* = \arg\max_{q_k \in \mathcal{Q}} \delta(q_k) \tag{9}$$

Only $q^*$ is retained, while the remaining candidates are discarded. The resulting pair $(p, q^*)$ is added to the training set $\mathcal{D}_{\text{hard}}$, which is used to fine-tune the problem generator $M_\theta$.

## 2.3 Difficulty-aware Graph Construction

This section extends our problem generation scaffolding (Section 2.1) to be difficulty-aware. In the baseline setup, the initial edge weights of the concept graph are determined primarily by the co-occurrence statistics of concepts within the same problems. Here, we further incorporate difficulty by modeling the hardness of concepts with respect to the difficulty levels of the problems in which they appear. Since each problem in the seed dataset (e.g., TACO; Li et al. (2023)) is annotated with human-curated difficulty labels, we leverage this information when constructing the concept graph for problem generation prompts. Specifically, beyond using co-occurrence counts as edge weights for random walk sampling, we also incorporate the average difficulty of all problems that involve both concepts connected by an edge. The new edge weights are defined as

$$w(u, v) = \log\left(\alpha \cdot \text{freq}(u, v) + (1 - \alpha) \cdot \text{diff}(u, v) + \epsilon\right)$$

$$\text{where } \text{diff}(u, v) = \frac{1}{|Q_{u,v}|} \sum_{q \in Q_{u,v}} d(q), \quad Q_{u,v} = \{ q \mid u \in q, \ v \in q \}. \tag{10}$$

Here, $\alpha$ is a hyperparameter that balances the contribution of co-occurrence frequency and difficulty and we set $\alpha = 0.2$ in our experiments. The constant $\epsilon$ is included to avoid taking the logarithm of zero. The set $Q_{u,v}$ consists of all problems containing both concepts $u$ and $v$; its cardinality is denoted by $|Q_{u,v}|$. Finally, $d(q)$ represents the human-annotated difficulty level of problem $q$, given as an integer from 1 to 5.

## 3 Experiments

In this section, we present the detail of our experiments. We first use our proposed difficulty measure method for data selection. Then we show the long CoT SFT results using datasets distilled from Qwen3-8B compared with previous strong baselines, and we show our generated datasets can also be effective when used in RL training. We further present an ablation study to investigate the effect of each role in our proposed method. Finally we have contamination analysis and statistics about our generated data.

## 3.1 Implementation Details

**Seed data:** We use TACO (Li et al., 2023) as seed data, which has human-annotated labels for difficult. TACO has 25.4K training samples and 1K test samples. Each problems is annotated with difficulty, test cases, and a list of topics. Samples in this dataset are collected from open-access sites where programmers share problems with each other, including Aizu, AtCoder, CodeChef, Codeforces, and LeetCode.

**Benchmarks** We use LiveCodeBench-V5 (Jain et al., 2025) and USACO as our evaluation benchmarks. We use LiveCodeBench-V5 for direct comparison with a strong baseline (Ahmad et al., 2025); USACO (Shi et al., 2024) is used because it is a representative code competition which contains difficult problems and has already been curated as benchmark for evaluation.

**Models:** We use Qwen3-8B as our teacher model in distillation experiments, as it is efficient and has competitive reasoning performance. We use Qwen2.5-Coder-7B-Instruct and Qwen3-8B-Base as our student model, respectively. For the RL experiments, we use Qwen2.5-7B-Instruct model as starting checkpoint for small-scale verification. We use Qwen2.5-14B-Instruct and GPT-4o as generators, as they can follow instructions relatively well compared to smaller models.

**Hyperparameters:** We use vLLM [1] as our inference framework for both distillation and evaluation experiments. We set temperature to 0.6 for all experiments. We set the batch size to 128 and the

---

[1] https://github.com/vllm-project/vllm

Table 2: Effect of different strata of synthetically generated coding problems on downstream performance. $\delta$ refers our estimated difficulty defined in Section 2.2 . Response length is determined based on responses generated by Qwen3-8B.

| Selection of problems | LiveCodeBench-V5 score | Avg. response length in tokens |
| --- | --- | --- |
| Random 3K | 36.29 | 11.9K |
| Highest $\delta$ 3K | 39.28 | 14.2K |
| Median $\delta$ 3K | 36.35 | 14.1K |
| Lowest $\delta$ 3K | 32.37 | 6.8K |
| Longest response 3K | 38.35 | 22.6K |

learning rate to 5e-5 for our SFT experiments, including the fine-tuning of the generator models. We use VeRL [2] for our RL experiments, and use 128 as the rollout batch size, 64 as the mini-batch size, and 16 as the rollout sample size. For all evaluation, we calculate averaged pass@1 across 16 runs.

## 3.2 USING ESTIMATED DIFFICULTY FOR DATA SELECTION

Before training the generator to produce difficult coding problems, we first need a trustworthy signal that can serve as a proxy for difficulty when gold labels are unavailable for generated problems. As mentioned above, we propose using $\delta$ we defined in Section 2.2 based on model responses. To verify the usefulness of this signal, we conduct a preliminary experiment that selects subsets of generated problems based on this signal for controlled comparison. We use our baseline graph random walking process to generate 50K problems using TACO as seed data. For each problem, we generate 8 responses and compute $\delta$. We then select 3K samples with the highest $\delta$, 3K with the lowest $\delta$, 3K with $\delta$ closest to 0.5, and an additional 3K randomly sampled for comparison. We also use response token length as another difficulty proxy and select 3K samples with the longest responses. Table 2 shows the results of using different selection methods and the performance of models trained on the selected problems, with 8 responses generated for each problem to ensure the scale and significance of our experiments. We observe that problems with the highest $\delta$ achieve the best performance, even surpassing those with the longest token responses, and using significant less tokens. We can also observe that for the problems with highest $\delta$, the token length is higher than problems with median and lowest $\delta$, which indicates there are some positive correlations between token length and $\delta$, but $\delta$ is still a more effective and efficient signal compared to response length.

## 3.3 TRAINED GENERATOR FOR DISTILLATION

We then use our trained generator to generate problems and leverage these problems to obtain responses from long chain-of-thought models (Qwen3-8B in our experiments) for training student models. In Table 3, we conduct a comprehensive comparison between previous Long CoT SFT datasets and our generated datasets on representative code reasoning benchmarks: LiveCodeBench-V5 and USACO. For our method (QueST), as described in Section 3.1, we use Qwen2.5-14B-Instruct to train a specialized generator under our reject fine-tuning and difficulty aware graph sampled prompt, 20K and 100K represents the training data size. "7B" means we trained from Qwen2.5-Coder-7B-Instruct, and "8B" means we trained from Qwen3-8B-Base.

Among the comparison group, OCR-Qwen-7B-Instruct Ahmad et al. (2025) stands out as the strongest competitor, leveraging DeepSeek-R1 as the teacher model and generating up to 32 responses for each of the 28K humanwritten coding problems. To ensure a fair comparison, we re-implement the OCR method using Qwen3-8B-Base as the student model and generate 4 responses per problem (yielding a total of 112K examples) using Qwen3-235B-A22B as the teacher. Even under these conditions, QueST-100K-8B outperforms OCR-8B across both benchmarks.

## 3.4 REINFORCEMENT LEARNING

Our generated data can also be used for RLVR (Reinforcement Learning with Verifiable Reward). We use majority voting results produced by Qwen3-8B as pseudo output labels for each test case

---

[2]https://github.com/volcengine/verl

Table 3: Performance on LiveCodeBench-V5 and USACO. Note: In our method, we only use Qwen3-8B as teacher model for 7B model, and Qwen3-235B-A22B as teacher model for 8B model. The content in brackets represents the generator models used for problem generation (GPT-4o for MathScale), in our methods, we use our trained Qwen2.5-14B-Instruct as generator. 20K and 100K means the number of training samples.

| Model | LiveCodeBench-V5 | | | | USACO | | | |
|---|---|---|---|---|---|---|---|---|
| | Easy | Medium | Hard | Avg. | Easy | Medium | Hard | Avg. |
| **Upper Bound (Teacher Models)** | | | | | | | | |
| DeepSeek-R1 | 98.5 | 79.8 | 37.4 | 65.6 | - | - | - | - |
| Qwen3-8B | 94.0 | 74.1 | 28.9 | 58.7 | 58.5 | 42.8 | 22.3 | 43.5 |
| **Baselines** | | | | | | | | |
| OpenThinker-7B (114K) | 80.6 | 16.9 | 1.6 | 25.5 | 11.0 | 2.1 | 0.0 | 5.0 |
| R1-Distill-Qwen-7B (800K) | 86.6 | 43.8 | 7.0 | 38.0 | 22.9 | 9.7 | 3.8 | 13.4 |
| OlympicCoder-7B (100K) | 82.1 | 49.4 | 12.2 | 40.9 | 31.4 | 12.5 | 1.3 | 17.0 |
| OCR-Qwen-7B-Instruct (700K) | 95.4 | 64.0 | 18.0 | 51.3 | 41.5 | 26.0 | 7.5 | 27.2 |
| MathScale-20K-7B (GPT-4o) | 82.8 | 36.6 | 8.2 | 34.9 | 28.0 | 15.6 | 1.3 | 16.7 |
| **Our Method** | | | | | | | | |
| QueST-20K-7B | 84.9 | 41.4 | 10.4 | 37.9 | 30.5 | 16.7 | 6.2 | 19.4 |
| QueST-100K-7B | 87.8 | 50.8 | 14.6 | 43.3 | 31.4 | 25.0 | 10.0 | 23.5 |
| OCR-8B (112K, Our Impl.) | 96.0 | 70.2 | 26.2 | 56.5 | 54.5 | 40.5 | 22.9 | 41.3 |
| QueST-100K-8B (100K, Ours) | 97.1 | 74.8 | 28.4 | 59.4 | 55.9 | 44.0 | 24.7 | 43.5 |
| QueST-8B (212K, Ours) | **97.6** | **81.0** | **36.6** | **65.2** | **65.5** | **48.6** | **28.7** | **49.9** |

Table 4: RL results on LiveCodeBench-V5

| Model | LiveCodeBench-V5 | | | |
|---|---|---|---|---|
| | Easy | Medium | Hard | Avg. |
| Qwen2.5-7B-Instruct | 47.4 | 8.4 | 0.1 | 14.3 |
| Qwen2.5-7B-Instruct TACO RL | 56.7 | 10.8 | 1.1 | 17.3 |
| Qwen2.5-7B-Instruct Baseline RL | 56.0 | 9.6 | 3.2 | 17.6 |
| Qwen2.5-7B-Instruct QueST RL | 56.4 | 9.6 | 4.8 | 18.6 |

of each generated problem. Since our generated test cases are not guaranteed to be valid, we filter out test cases where over half of the outputs are none (indicating failed execution for generated solutions), then keep the remainder for RLVR. We use the GRPO (Shao et al., 2024) algorithm to train Qwen2.5-7B-Instruct on 12K problems sampled from TACO, 6K data from our baseline synthetic method (mathscale) (Tang et al., 2024), and 6K data from QueST. We report our results in Table 4, which shows effectiveness of our proposed method.

We report the training reward curve during the training process in Figure 2. It shows that the model trained on TACO datasets gains the highest reward score during the whole training stage, our baseline synthetic method gains a lower score, and the model trained on a dataset generated by the QueST method gains the lowest score. The training reward can serve as a proxy of the inverse difficulty of these three different datasets.

## 3.5 ABLATION STUDY

We conducted an ablation study for fair comparison across different settings, as shown in Table 5. In the first two rows of the table, we examine whether using difficulty-aware graph random walking improves performance when using GPT-4o as the generator. The results demonstrate that the difficulty-aware graph achieves clear improvement. In the third, fourth and fifth rows, we compare performance when only using Qwen2.5-14B-Instruct (Baseline), with the difficulty-aware graph but Qwen2.5-14B-Instruct without further training and Qwen2.5-14B-Instruct under our rejection fine-tuning method (QueST). The results show that when using difficulty-aware random sampling

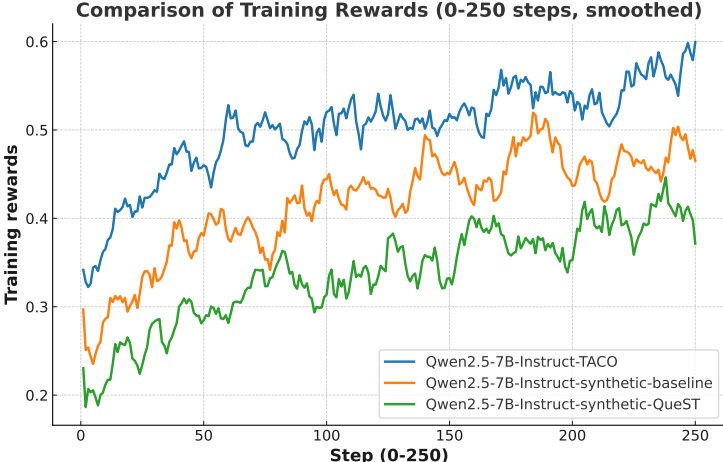

Figure 2: Training rewards comparison in the training process of RL under different datasets.

prompts, our fine-tuned generator can bring better performance than the model without using our fine-tuning method, and better than baseline. Therefore, Table 5 indicate both difficulty-aware sampling and rejection fine-tuning have positive effect and lead to generating difficult problems.

Table 5: Ablation study on LiveCodeBenchV5. "Baseline" here represents the our baseline problem generation pipeline (Tang et al., 2024) which we discussed in Section 2.1. Here we generate 20K questions for all settings to fair comparison, and the base model we used to train is Qwen2.5-Coder-7B-Instruct. "RFT" is abbreviation of our rejection fine-tuning method.

| Methods | LiveCodeBench | | | |
| --- | --- | --- | --- | --- |
| | Easy | Medium | Hard | Avg. |
| Problem Generator: GPT-4o | | | | |
| Baseline | 82.8 | 36.6 | 8.2 | 34.9 |
| Baseline w/ difficulty-aware graph | 83.6 | 41.1 | 10.9 | 37.5 |
| Problem Generator Qwen2.5-14B-Instruct | | | | |
| Baseline | 81.7 | 32.6 | 6.1 | 32.8 |
| Baseline w/ difficulty-aware graph | 85.0 | 39.2 | 8.0 | 36.1 |
| Baseline w/ difficulty-aware graph w/ RFT (QueST) | 84.9 | 41.4 | 10.4 | 37.9 |

## 3.6 ADDITIONAL ANALYSIS

We visualize and compare the 25 most sampled knowledge points with and without difficulty-aware sampling in Appendix Figure 4. The figure shows that knowledge points sampled more frequently by naive sampling than by difficulty-aware sampling tend to be more common overall, while knowledge points sampled less frequently by naive sampling tend to be less common. In other words, difficulty-aware sampling upweights infrequent knowledge points and downweights frequent knowledge points compared to naive sampling. The infrequent knowledge points are visualized in the left figure and are generally more difficult, including topics such as the "knapsack problem", "Optimal Play Strategies", and "prime factorization", compared to the basic concepts shown in the right figure.

We also conduct a case study on generated problems from both original model and model trained by QueST framework in Appendix Table 6. It shows that the problem generated by our trained model requires more complex operations and more knowledge compared the question generated by original model.

We conduct contamination detection experiments on our generated datasets to exclude the effects of data contamination on benchmark performance. Specifically, we compute token-based 50-gram Jaccard similarity scores and the scores across all datasets and benchmarks we used are 0 which indicates there is no contamination in our generated data.

We also conducted correlation analysis of our computed difficulty estimation with human labeled difficulty ratings in TACO dataset. We transform difficulty levels of TACO to 1-5 scores as rating levels, corresponding to easy, medium, medium hard, hard, and very hard, respectively. We report the distribution and mean scores of average majority voting rate for each problems in each human labeled rating in Figure 3. And it shows a trend that when rating become larger (more difficult), the mean of average majority voting rate become lower.

## 4 RELATED WORK

### 4.1 SYNTHETIC DATA FOR LANGUAGE MODELS

Synthetic data has been widely used in training language models. Previous works have mainly focused on using small sets of seed data and leveraging LLMs to augment them and generate larger datasets. Some works (Honovich et al., 2023; Li et al., 2024a; Toshniwal et al., 2025; Wang et al., 2023b; Tang et al., 2024) focus on sampling seed data as in-context learning exemplars to generate new ones. Ge et al. (2025) proposed using personas to augment previous in-context learning synthetic data generation methods. Xu et al. (2024); Luo et al. (2025a); Hu et al. (2025) focus on augmenting existing samples to create more complex ones. Some methods have also explored how to generate synthetic data from scratch (Li et al., 2024b; Xu et al., 2025). More recently, Qin et al. (2025) investigated whether synthetic data follows similar scaling laws as real data. Shah et al. (2025); Kaur et al. (2025); Didolkar et al. (2024) extract skills and generate new problems. PromptCoT (Zhao et al., 2025) also generates challenging problems based on mathematical concepts and rationale. Tong et al. (2024) also proposed a difficulty-aware method but focuses on synthetic responses for challenging problems. Liang et al. (2025) extract concepts from failure cases and synthesize new problems during RL training. Additionally, there is research focused on leveraging pretraining or web data to generate reasoning data in general domains (Yuan et al., 2025; Yue et al., 2024). Our QueST framework focuses on a new perspective that aims to train a difficulty-aware generator to generate difficult problems.

### 4.2 CODE REASONING

Code reasoning is an important capability of large language models. The reasoning ability of language models can be enhanced using chain-of-thought (Wei et al., 2022), RLVR (OpenAI, 2024; Guo et al., 2025; Lambert et al., 2025), and self-consistency (Wang et al., 2023a), in math (Hendrycks et al., 2021) and code (Jain et al., 2025; Shi et al., 2024) domains. Muennighoff et al. (2025) and Ye et al. (2025) focus on manually curating small-scale reasoning problems, which is sufficient to boost models' reasoning ability. More recently, Face (2025), Ahmad et al. (2025), and Guha et al. (2025) have developed large-scale distillation methods from reasoning models to obtain high-quality long CoT SFT datasets that can be used to train student models effectively. Nvidia et al. (2024) curate reasoning datasets throughout the entire training pipeline. Li et al. (2025) introduce an innovative paradigm that transforms traditional code reasoning tasks from their original format into a "given code + test cases / input-output prediction" structure. Complementing these supervised learning approaches, Luo et al. (2025b) demonstrate the effectiveness of reinforcement learning techniques applied to verified code reasoning problems. However, how to generate difficult synthetic coding problems and use them for training remains relatively underexplored.

## 5 CONCLUSION

In this paper, we propose a method for generating difficult code problems at scale. Specifically, we investigated a pipeline that uses majority voting to compute a proxy of difficulty and employs this as a signal for rejection fine-tuning of the problem generator, and combined it with novel difficulty-aware graph sampling prompts. This enables the trained generator to produce challenging problems at scale. We then use these generated problems for supervised fine-tuning (SFT) and reinforcement learning (RL) to verify their effectiveness. As a novel synthetic data generation method, we compared our

approach with previous baselines at similar scales on code reasoning benchmarks and show that our method achieves better performance even when using less SFT data, particularly for hard problems.

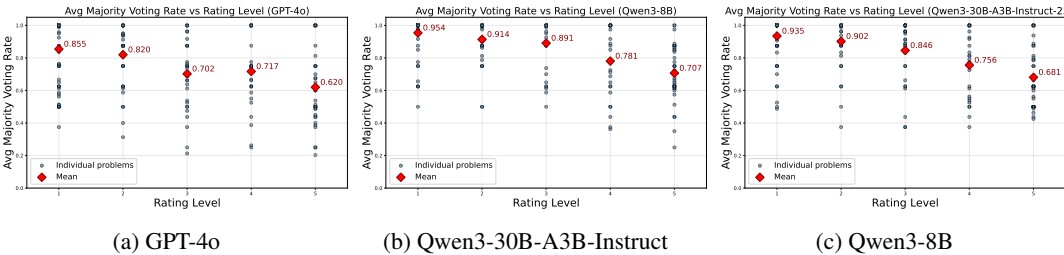

|  (a) GPT-4o | (b) Qwen3-30B-A3B-Instruct | (c) Qwen3-8B |

Figure 3: Avg Majority Voting Rate vs Rating Level (Difficulty) in TACO.

## LIMITATIONS AND FUTURE WORK

Although our method shows promise for rejection fine-tuning a generator, we still face limitations as the generator hasn't been trained using RL. One primary reason is that our current difficulty calculation is computationally expensive and challenging to implement in real-time to provide difficulty rewards in an RL pipeline, considering that we need to generate 8 responses and 20 test cases for each problem on the fly, execute them, and generate $K$ problems for each prompt. In future work, it would be worthwhile to explore methods that can provide rewards in real time, such as directly training a reward model to predict difficulty, or investigating other efficient approaches.

## REPRODUCIBILITY STATEMENT

To help community reproduce our work, we described details of implementation in Section 3.1, which reports the details of data, benchmark, models, and hyperparameters we use in our experiments. We also report the framework we use for training and inference. In Appendix Figure 5 6 7, we report the prompt template we use.

## ETHICS STATEMENT

In the paper, all the data we use is open-sourced. TACO (Li et al., 2023) has Apache-2.0 license. LiveCodeBench (Jain et al., 2025) and USACO (Shi et al., 2024) are collected from open part of common competition websites.

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

# A APPENDIX

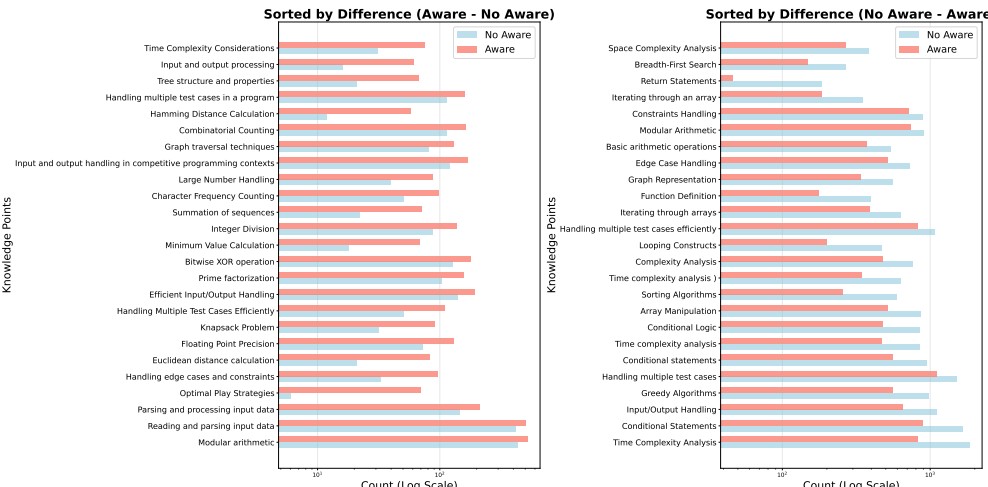

Figure 4: Examples of knowledge points where the count is increased (left) or decreased (right) when using difficulty-aware sampling.

## A.1 USAGE OF LLM

We used LLMs to polish the writing and assist in the implementation of parts of the codebase. All the methodologies, ideas, conceptual contributions, experiment designs, analyses were developed solely by the authors.

Table 6: Case Study: Topic and knowledge points that were provided to the generator, and two generated questions. Left is generated by our trained model, right is generated by original Qwen2.5-14B-Instruct.

| Topic | String Analysis | |
| --- | --- | --- |
| Knowledge points | 1. Hamming Distance Calculation; 2. Time Complexity Optimization; 3. Conditional Logic and State Management; 4. Event Handling and Simulation; 5. Priority Queue | |
| Problem | You are given a string s of length n consisting of binary characters ('0' and '1'). You need to perform several queries on this string. Each query can be one of the following types:
1. set(l, r, c): Set all characters in the substring s[l:r+1] to the character c (either '0' or '1').
2. flip(l, r): Flip all characters in the substring s[l:r+1] (change '0' to '1' and '1' to '0').
3. hamming(l1, r1, l2, r2): Calculate the Hamming distance between the substrings s[l1:r1+1] and s[l2:r2+1].

Your task is to handle these queries efficiently.

Input:
- The first line contains the string s of length n ($1 \leq n \leq 200{,}000$).
- The second line contains an integer q ($1 \leq q \leq 400{,}000$) representing the number of queries.
- The next q lines describe the queries. Each query is in one of the above formats.
- For each hamming query, output the Hamming distance between the specified substrings. | In a messaging system, messages are sent as strings of binary digits. Sometimes, due to technical issues, certain bits in the messages might get flipped (0 becomes 1, and 1 becomes 0). To ensure the integrity of the messages, a special protocol is used where each message is sent alongside its "flipped" version. Upon receiving both versions, the receiver calculates the Hamming distance between the original and the flipped message to detect any errors.

Given two strings s and t of equal length representing the original and flipped versions of a message, your task is to write a function that calculates the Hamming distance between them. The Hamming distance is defined as the number of positions at which the corresponding bits are different. Additionally, the function should handle up to $10^5$ queries efficiently, optimizing for time complexity. |

Act as a Programming Teacher and create a new question and its solution based on the provided topics and knowledge points. Ensure that the created questions:
1. Adhere to the provided topics.
2. Necessitate the combined use of the associated knowledge points.

# Example
Topics:
1. Graph Theory
2. Combinatorics
Knowledge Points:
1. Graph Coloring
2. Connected Components
3. Dynamic Programming
4. Modular Arithmetic
Created Question:
You are given a grid, consisting of $2$ rows and $n$ columns. Each cell of this grid should be colored either black or white.
Two cells are considered neighbours if they have a common border and share the same color. Two cells $A$ and $B$ belong to the same component if they are neighbours, or if there is a neighbour of $A$ that belongs to the same component with $B$.
Let's call some bicoloring beautiful if it has exactly $k$ components.
Count the number of beautiful bicolorings. The number can be big enough, so print the answer modulo $998244353$.

-----Input-----
The only line contains two integers $n$ and $k$ ($1 \le n \le 1000$, $1 \le k \le 2n$) — the number of columns in a grid and the number of components required.

-----Output-----
Print a single integer — the number of beautiful bicolorings modulo $998244353$.

-----Examples-----
Input
3 4
Output
12
Input
4 1
Output
2
Input
1 2
Output
2

Topics:
1. String Manipulation
Knowledge Points:
1.   Understanding and manipulating string data structures
2.   Dynamic Programming

Try to create a question for the last one.  Structure your response as:
Created Question:
<Question>

Figure 5: 1-shot prompt example for problem generation. It is simplified for visualization, in real prompt, we have 8-shot for in-context learning.

You are an expert programmer. Your task is to write some test cases to the programming problems to help verify the expected program solutions. You only need to give me the inputs in the required format. Now, let me introduce the details to you:

## Program Format

You will be given programming problems that accept standard input-output stream. As a result, the test case inputs should contain only the inputs text stream.

## Response Format

You should return me the test case inputs in `json_object` format. You need to generate **20** groups of test case inputs, and each key field is named as `test_case_i`, where `i` is the index of the test case. The value of each key is the test case inputs in the required format.

## Example for Standard Input-Output Stream

#### Programming Problem

Polycarp has $n$ different binary words. A word called binary if it contains only characters '0' and '1'. For example, these words are binary: "0001", "11", "0" and "0011100".

Polycarp wants to offer his set of $n$ binary words to play a game "words". In this game, players name words and each next word (starting from the second) must start with the last character of the previous word. The first word can be any. For example, these sequence of words can be named during the game: "0101", "1", "10", "00", "00001".

Word reversal is the operation of reversing the order of the characters. For example, the word "0011" after the reversal becomes "1110", the word "11010" after the reversal becomes "01011".

Probably, Polycarp has such a set of words that there is no way to put them in the order correspondent to the game rules. In this situation, he wants to reverse some words from his set so that: the final set of $n$ words still contains different words (i.e. all words are unique); there is a way to put all words of the final set of words in the order so that the final sequence of $n$ words is consistent with the game rules.

Polycarp wants to reverse minimal number of words. Please, help him.

-----Input-----

The first line of the input contains one integer $t$ ($1 \le t \le 10^4$) — the number of test cases in the input. Then $t$ test cases follow.

The first line of a test case contains one integer $n$ ($1 \le n \le 2\cdot10^5$) — the number of words in the Polycarp's set. Next $n$ lines contain these words. All of $n$ words aren't empty and contains only characters '0' and '1'. The sum of word lengths doesn't exceed $4\cdot10^6$. All words are different.

Guaranteed, that the sum of $n$ for all test cases in the input doesn't exceed $2\cdot10^5$. Also, guaranteed that the sum of word lengths for all test cases in the input doesn't exceed $4\cdot10^6$.

-----Output-----

Print answer for all of $t$ test cases in the order they appear.

If there is no answer for the test case, print -1. Otherwise, the first line of the output should contain $k$ ($0 \le k \le n$) — the minimal number of words in the set which should be reversed. The second line of the output should contain $k$ distinct integers — the indexes of the words in the set which should be reversed. Words are numerated from $1$ to $n$ in the order they appear. If $k=0$ you can skip this line (or you can print an empty line). If there are many answers you can print any of them.

-----Example-----
Input
4
4
0001
1000
0011
0111
3
010
101
0
2
00000
00001
4
01
001
0001
00001

Output
1
3
-1
0

2
1 2

#### Response

```
{
  "test_case_0": "3\n3\n101\n110\n011\n2\n01\n10\n4\n0001\n1000\n0011\n0111",
  "test_case_1": "2\n2\n01\n10\n3\n000\n111\n110",
  ...
}
```

## Get Started

Note that in the above example, I omit some test case inputs. You should return **20** groups of inputs to me in `json_object` format.

#### Programming Problem

{problem}

#### Response

Figure 6: 1-shot example prompt for testcase generation.

```
Act as a Programming Teacher and analyze the provided question. Start by identifying 1 or 2 general topics
that a student is being assessed on. Structure your response as:
"Topics:
1. <Topic 1>
2. <Topic 2>"

Next, highlight 1 to 5 specific knowledge points that the question evaluates. Structure your response as:
"Specific Knowledge Points:
1. <Knowledge Point 1>
2. <Knowledge Point 2>
3. <Knowledge Point 3>
4. <Knowledge Point 4>
5. <Knowledge Point 5>"

The topics and specific knowledge points should be terms that are concise and commonly used in academia
or industry.

### Provided question:
{{ question }}

### Analysis:
```

Figure 7: Prompt demonstration for concept extraction.

