# OpenReview forum: "QueST: Incentivizing LLMs to Generate Difficult Problems"
_ICLR.cc/2026/Conference — Submitted to ICLR 2026_

### Official Review · Reviewer_HaRZ · 2025-10-20

**Soundness:** 2
**Presentation:** 3
**Contribution:** 2
**Rating:** 2
**Confidence:** 4

**Summary:**

Large language models show strong capabilities in complex reasoning tasks. Training with difficult problems can further improve the model's performance. Based on this, the authors proposed a data synthesis method to generate more challenging problems. Specifically, they combined difficulty-aware graph sampling for prompts and difficulty-aware rejection fine-tuning to create high-difficulty training data. Training the model with this synthetic data led to some improvement in its performance.

**Strengths:**

+ The authors generated a set of training data related to coding tasks, totaling 100,000 examples.

+ The authors trained the model using the synthetic data, which led to improvements.

**Weaknesses:**

+ In Section 3.1 of the paper, the data synthesis process requires a large number of external models for assistance, but the authors do not explain the strategy for selecting these models, which makes it unconvincing.

+ The authors cannot explain the reason for the low accuracy in the synthetic problems. On one hand, it may be due to higher difficulty, and on the other hand, it may be because the answers contain errors.

+ How well does the problem generator perform on coding tasks? The paper does not explain this. If the trained model cannot outperform the problem generator, then the synthetic data is meaningless, especially since the authors claim in the abstract that "QueST pushes the boundaries of reasoning abilities in large language models."

**Questions:**

+ Please explain the selection strategy for the various models in Section 3.1.

+ Please provide the performance of the problem generator on coding tasks.

+ Why not use OCR-Qwen-7B-Instruct as the backbone model? It has stronger reasoning capabilities.

+ How does QueST perform compared to TACO when the data volume is the same? If QueST cannot significantly outperform TACO, it cannot be concluded that QueST's problems are more difficult or of higher quality.

+ The performance of the three models with RL in Table 4 shows no significant improvement and is unstable. Does this indicate that the quality of QueST's data is normal?

---

> ### Author Response · Authors · 2025-11-22
>
> Dear reviewer HaRZ
>
> We appreciate your valuable review, regarding the weakness and questions you mentioned, we reply as below:
>
> > In Section 3.1 of the paper, the data synthesis process requires a large number of external models for assistance, but the authors do not explain the strategy for selecting these models, which makes it unconvincing.
>
> Regarding the selection strategy in Section 3.1, we wish to clarify that our model choices were not arbitrary but driven by three constraints: representativeness, computational feasibility, and baseline alignment.
>
> First, for problem generation, we used a widely-used representative closed-source model (GPT-4o) with a largest open-weight model our computational resources could support for fine-tuning (Qwen2.5-14B-Instruct) to ensure high-quality synthesis. Second, our student models (Qwen-2.5-Coder and Qwen3-8B-Base) were specifically chosen to align with the settings in OpenCodeReasoning [1], allowing for direct comparison, while Qwen-2.5-7B-Instruct was selected for RL as a standard small-model baseline. Finally, Qwen3-8B served as the teacher model due to its strong reasoning capabilities relative to its small size. We have added this rationale to the paper to improve clarity.
>
> > The authors cannot explain the reason for the low accuracy in the synthetic problems
>
> We respectfully seek clarification regarding the mention of 'low accuracy in the synthetic problems,' as we did not explicitly report accuracy metrics for the synthetic dataset in the paper.
>
> If this comment refers to the low reward curves observed during RL (Figure 2) or the low consistency/majority voting rates used as a proxy for difficulty, we acknowledge that synthetic data may inherently contain some noise or errors. However, we argue that the distinction between 'erroneous' and 'difficult' is best validated by the training outcomes. If the synthetic problems were merely erroneous rather than genuinely challenging, using them for training would likely degrade model performance.
> Instead, our method achieves performance superior to previous methods in fair comparisons in Table A, and achieves state-of-the-art performance among 8B models in LCB when further scaling up in our latest results Table 3 in revised paper, suggesting the data provides a valid learning signal. Furthermore, to validate our difficulty measurement, we demonstrated positive Pearson and Spearman correlations between human-labeled difficulty in the TACO dataset and our model’s output consistency in Table B. This supports the hypothesis that our synthetic problems effectively capture difficulty.
>
> **Table A, Comparison with TACO using Qwen-235B-A22B as teacher model. Pass@1 in LiveCodeBenchV5**
>
> | Student Model | TACO | Ours |
> | :--- | :--- | :--- |
> | Qwen3-8B-Base | 0.4608 | **0.4998** |
>
> **New results in Table 3 in revised paper, 'Our Impl.' means using same strong teacher model (Qwen-235B-A22B) with ours**
>
> | Method | LiveCodeBenchV5 | USACO |
> | :--- | :--- | :--- |
> | OCR-8B (112K, Our Impl.) | 56.5 |  41.3 |
> | QueST-100K-8B (100K, Ours) | 59.4 | 43.5 |
> | QueST-8B (212K, Ours) | **65.2** | **49.9** |
>
> **Table B**: Correlations with human rated difficulty (From TACO, we transform difficulty to 1-5 scores.)
>
> | Correlations/Models | GPT-4o | Qwen3-8B | Qwen3-30B-A3B-Instruct-2507 |
> | :--- | :--- | :--- | :--- |
> | Spearman | 0.38, p-value=1.9e-8 | 0.54, p-value=9e-15 | 0.51, p-value=9e-15 |
> | Pearson | 0.39, p-value=1.5e-8 | 0.51, p-value=8e-17 | 0.49, p-value=6e-14 |
>
> > How well does the problem generator perform on coding tasks? The paper does not explain this. If the trained model cannot outperform the problem generator, then the synthetic data is meaningless
>
> We would like to clarify that the models used for problem generation are standard instruction-tuned models rather than specialized reasoning models; consequently, their baseline performance is relatively limited. We evaluated the performance of the generator models on LiveCodeBench as follows:
>
> | Generator Model | LiveCodeBench-V5 |
> | :--- | :--- |
> | Qwen2.5-14B-Instruct | 20.3 |
> | GPT-4o | 29.5 |
>
> These scores are notably lower than those achieved by our trained solver model. This performance gap demonstrates that our model successfully transcends the capabilities of the generator, validating the effectiveness of the synthetic data.

---

> ### Author Response · Authors · 2025-11-22
>
> > Why not use OCR-Qwen-7B-Instruct as the backbone model? It has stronger reasoning capabilities.
>
> Regarding the suggestion to use OCR-Qwen-7B-Instruct as the backbone: It is important to note that this model is the output of a specific synthetic SFT data creation method [1]. Since our objective is to compare the efficacy of our data synthesis method against theirs, we must begin from the same baseline. Both methods utilize Qwen-2.5-7B-Coder as the foundation. Using their fine-tuned model as our backbone would contaminate the experimental setting, making it impossible to isolate the performance gains attributable solely to our data generation approach.
>
> > How does QueST perform compared to TACO when the data volume is the same? If QueST cannot significantly outperform TACO, it cannot be concluded that QueST's problems are more difficult or of higher quality
>
> We address this concern directly in Table A (See above first comment), where we present a controlled comparison between QueST and TACO using identical data volumes. The results demonstrate that QueST outperforms TACO under this setting. This empirical evidence supports the conclusion that the performance improvements are driven by the superior quality and complexity of the QueST problems, rather than data scale.
>
> > Does this indicate that the quality of QueST's data is normal?
>
> While the definition of 'normal' in this context is slightly ambiguous, we interpret this as a question regarding whether our data quality is merely average. Our results suggest otherwise. Our QueST demonstrates improvements in both SFT and RL performance. In SFT settings, we have shown that we can achieve better results in Table A (See above first comment) when data size is the same, and achieve state-of-the-art when further scaling up in Table 3 (See above comment or in our revised paper) in the revised paper. In RL setting,  its advantage becomes significantly more pronounced on the 'Hard' subset. This specific improvement aligns perfectly with our method's objective to tackle complex reasoning tasks, indicating that the data is of high quality and difficulty rather than being standard or 'normal'.
>
>
> [1] OpenCodeReasoning: Advancing Data Distillation for Competitive Coding. Ahmad et al.

---

> ### Comment · Reviewer_HaRZ · 2025-11-27
>
> Thanks for the detailed response.
>
> I still have several concerns about whether the data generated from Qwen2.5-14B-Instruct can be utilized to improve itself. And what is the performance gap between training with synthetic data and direct distillation?

---

> ### Author Response · Authors · 2025-11-27
>
> Dear reviewer HaRZ,
>
> In terms of coding capabilities, yes.
>
> As shown in the results below, QueST-7B (student: Qwen2.5-7B-Coder) achieves 43.3 on LCBv5, substantially outperforming Qwen2.5-14B-Instruct (20.3). Notably, Qwen2.5-14B-Instruct is a strictly stronger base model than Qwen2.5-7B-Coder (LCBv5: 20.3 vs. 16.5). Therefore, it is reasonable to expect that using a stronger student (e.g., Qwen2.5-14B-Instruct) would yield results that exceed the current QueST-7B score of 43.3, and far surpass its original baseline (20.3).
>
> For the primary experiments, we use Qwen3-8B-Base as the student. Our goal is to evaluate whether QueST-8B can surpass the performance of Qwen3-8B, since both use the same teacher model (Qwen3-235B) and the same student architecture, but Qwen team did not release the coding-problem dataset they used for distillation. Using our newly generated, high-difficulty code problems, QueST-8B achieves significantly stronger performance than Qwen3-8B, confirming that our synthetic-data pipeline provides clear and consistent improvements.
>
> | Models   | LCBv5 |
> | :--- | :--- |
> | Qwen2.5-14B-Instruct | 20.3 |
> | Qwen2.5-7B-Instruct |  14.3 |
> | Qwen2.5-7B-Coder |  16.5 |
> | Quest-7B (Qwen2.5-7B-Coder as student, Qwen3-8B as Teacher) | **43.3** |
> | Qwen3-8B | 58.7 |
> | Quest-8B (Qwen3-8B-Base as student, Qwen3-235B-A22B as teacher) | **65.2** |
>
>
>
> Could you clarify your second question "And what is the performance gap between training with synthetic data and direct distillation?"?

---

### Official Review · Reviewer_jBTM · 2025-10-26

**Soundness:** 2
**Presentation:** 3
**Contribution:** 2
**Rating:** 4
**Confidence:** 4

**Summary:**

This work presents a difficulty-aware framework for generating coding problems, integrating difficulty-guided graph sampling with rejection-based fine-tuning. Additionally, the authors release a 100K dataset, which contributes valuable resources for future research and community use.

**Strengths:**

1. Difficulty Estimation via Self-Consistency: The authors estimate problem difficulty using self-consistency across multiple model outputs.
2. Difficulty-Guided Sampling: For each prompt, multiple candidate problems are generated, and only the most difficult one (based on the proposed difficulty metric) is retained for training.
3. Instead of letting the model generate simple problems repeatedly, the method continuously selects and trains on the most challenging problems, thus enhancing the generator’s reasoning and problem-design ability. A “difficulty label” is introduced such that concept pairs frequently co-occurring in hard problems are more likely to be sampled together during generation.

**Weaknesses:**

1. The paper does not introduce a fundamentally new data synthesis approach but rather extends MathScale with heuristic sampling improvements based on self-consistency. Such heuristics are intuitive but may not lead to substantial long-term impact.

2. The proposed method for measuring problem difficulty mainly relies on self-consistency within rollouts, which appears heuristic and lacks deeper theoretical justification to confirm its validity.

3.The synthetic data do not show clear superiority in experimental results: for example, in Table 3, QueST-100K-7B underperforms OCR-Qwen-7B-Instruct. The best results rely on Qwen3-8B, which also raises concerns about fairness and comparability due to missing baselines.

4. Although the method does not require human annotation, it still depends on TACO as the seed dataset, meaning it is not a fully self-synthesized approach.

**Questions:**

In lines 118–120, how are semantically similar but different concepts handled? Given that generated concepts may exhibit such redundancy, could this affect sampling consistency or downstream results?

---

> ### Author Response · Authors · 2025-11-22
>
> Dear reviewer jBTM
>
> We appreciate your valuable review, regarding the weakness and questions you mentioned, we reply as below:
>
> Weakness 1:
>
> > The paper does not introduce a fundamentally new data synthesis approach but rather extends MathScale
>
> We wish to clarify that MathScale serves only as a scaffold within our pipeline; it is not the core focus of our work. Our primary contribution is a novel method for **difficulty-aware** problems synthesis. To the best of our knowledge, we are the first to train generators specifically optimized to control and elevate problem difficulty, which we demonstrate is critical for improving downstream model performance. Furthermore, our proposed rejected fine-tuning method is model-agnostic and can be readily adapted to other synthetic data pipelines.
>
> Weakness 2:
> > The proposed method for measuring problem difficulty mainly relies on self-consistency within rollouts, which appears heuristic and lacks deeper theoretical justification to confirm its validity.
>
> We address this concern from three perspectives: theoretical grounding in prior literature, empirical alignment with human judgment, and downstream task effectiveness.
>
> **Prior Literature**: As noted in Section 2.2, our approach builds upon the seminal work of Self-Consistent CoT (Wang et al.), which finds that self-consistency is highly correlated with accuracy, which reflects the uncertainty of the models, and also the difficulty of the problem.
>
> **Alignment with Human Judgment**: To further validate this metric, we have provided new empirical evidence in the added Table B. These results demonstrate a strong positive correlation between our self-consistency scores and human-labeled difficulty levels, confirming that our metric reflects intrinsic problem hardness.
>
> **Downstream Effectiveness**: Beyond correlations, we believe the most critical validation of a metric is its utility. We demonstrate that using this difficulty measure to curate training data leads to superior downstream performance. Specifically, our method achieves higher performance compared to baselines using the same volume of training data in Table A (3.9 pass@1 gain), and our latest updated results in Table 3 (see in revised paper or below next comment) which gain state-of-the-art performance in LCB benchmark among 8B models, demonstrating the practical validity of the proposed metric.
>
> Weakness 3:
> > The synthetic data do not show clear superiority in experimental results: for example, in Table 3, QueST-100K-7B underperforms OCR-Qwen-7B-Instruct. The best results rely on Qwen3-8B, which also raises concerns about fairness and comparability due to missing baselines.’
>
> Regarding the performance gap between QueST-100K-7B and OCR-Qwen-7B-Instruct in Table 3 (see our revised paper or below next comment), the initial comparison was influenced by a disparity in the teacher models used for distillation. Specifically, the OCR baseline benefited from the much larger DeepSeek-R1 as a teacher, whereas our reported model relied on the smaller Qwen3-8B.
> To address the concern regarding fairness and to provide a rigorous comparison, we conducted a new set of controlled experiments. We employed the stronger teacher model Qwen3-235B-A22B for generating response for both the OCR baseline problem data and our synthetic data. We re-implement the OCR method using
> Qwen3-8B-Base as the student model and generate 4 responses per problem (yielding a total of
> 112K examples). Even
> under these conditions, QueST-100K-8B outperforms OCR-8B across both benchmarks.  Furthermore, by combining the 112K organic examples with our 100K generated examples, we achieve even stronger results (QueST-8B-212K), attaining performance on par with the much larger DeepSeek-R1-671B.
>
> Weakness 4:
> > Although the method does not require human annotation, it still depends on TACO as the seed dataset, meaning it is not a fully self-synthesized approach.
>
> We acknowledge that our method utilizes TACO as a seed dataset. However, utilizing seed data to bootstrap the generation of complex queries is a standard and practical setting in current synthetic data research. Crucially, TACO is used solely to train the generator. Once trained, the generator is capable of synthesizing a theoretically unlimited amount of data without further human supervision. To demonstrate this scalability, we have already utilized our trained generator to synthesize 100k additional samples, proving that our approach goes well beyond the initial seed constraints.
>
> Questions:
> > In lines 118–120, how are semantically similar but different concepts handled? Given that generated concepts may exhibit such redundancy, could this affect sampling consistency or downstream results?
>
> Controlling diversity is an intriguing possibility for future work. It is likely that diversity affects downstream results, but our focus is on the aspect of task difficulty.

---

> ### Author Response · Authors · 2025-11-22
>
> **Table A, Comparison with TACO using Qwen-235B-A22B as teacher model. Pass@1 in LiveCodeBenchV5**
>
> | Student Model | TACO | Ours |
> | :--- | :--- | :--- |
> | Qwen3-8B-Base | 0.4608 | **0.4998** |
>
> In Table A, we use all TACO training data, and control same size when using our data for fair comparison.
>
> **Table B: Correlations with human-rated difficulty (From TACO, we transform difficulty levels to 1-5 scores, corresponding to easy, medium, medium hard, hard, and very hard, respectively.)**
>
>
>
> | Correlations/Models | GPT-4o | Qwen3-8B | Qwen3-30B-A3B-Instruct-2507 |
> | :--- | :--- | :--- | :--- |
> | Spearman | 0.38, p-value=1.9e-8 | 0.54, p-value=9e-15 | 0.51, p-value=9e-15 |
> | Pearson | 0.39, p-value=1.5e-8 | 0.51, p-value=8e-17 | 0.49, p-value=6e-14 |
>
>
>
> **Table 3 Full Results Comparison With Previous Baselines and Stronger Settings in Our implementations.**
> | Model | LCB-V5 Easy | LCB-V5 Medium | LCB-V5 Hard | LCB-V5 Avg. | USACO Easy | USACO Medium | USACO Hard | USACO Avg. |
> | :--- | :---: | :---: | :---: | :---: | :---: | :---: | :---: | :---: |
> | DeepSeek-R1-671B (Guo et al., 2025) | 98.5 | 79.8 | 37.4 | 65.6 | 72.5 | 54.6 | 34.3 | 56.2 |
> | Qwen3-8B (Yang et al., 2025) | 94.0 | 74.1 | 28.9 | 58.7 | 58.5 | 42.8 | 22.3 | 43.5 |
> | R1-0528-Qwen3-8B (Guo et al., 2025) | 94.4 | 73.5 | 27.7 | 58.1 | 57.0 | 33.6 | 17.2 | 38.5 |
> | OpenThinker-7B (114K) (Guha et al., 2025) | 80.6 | 16.9 | 1.6 | 25.5 | 11.0 | 2.1 | 0.0 | 5.0 |
> | R1-Distill-Qwen-7B (800K) (Guo et al., 2025) | 86.6 | 43.8 | 7.0 | 38.0 | 22.9 | 9.7 | 3.8 | 13.4 |
> | OlympicCoder-7B (100K) (Face, 2025) | 82.1 | 49.4 | 12.2 | 40.9 | 31.4 | 12.5 | 1.3 | 17.0 |
> | OCR-Qwen-7B-Instruct (700K) (Ahmad et al., 2025) | 95.4 | 64.0 | 18.0 | 51.3 | 41.5 | 26.0 | 7.5 | 27.2 |
> | OCR-8B (112K, Our Impl.) | 96.0 | 70.2 | 26.2 | 56.5 | 54.5 | 40.5 | 22.9 | 41.3 |
> | QueST-100K-8B (100K, Ours) | 97.1 | 74.8 | 28.4 | 59.4 | 55.9 | 44.0 | 24.7 | 43.5 |
> | **QueST-8B (212K, Ours)** | **97.6** | **81.0** | **36.6** | **65.2** | **65.5** | **48.6** | **28.7** | **49.9** |
>
>
> In Table 3, we employed the stronger teacher model Qwen3-235B-A22B for generating response for both the OCR baseline problem data and our synthetic data. We re-implement the OCR method using Qwen3-8B-Base as the student model and generate 4 responses per problem (yielding a total of 112K examples). Even under these conditions, QueST-100K-8B outperforms OCR-8B across both benchmarks. Furthermore, by combining the 112K organic examples with our 100K generated examples, we achieve even stronger results (QueST-8B-212K), attaining performance on par with the much larger DeepSeek-R1-671B.

---

### Official Review · Reviewer_CXvL · 2025-10-28

**Soundness:** 2
**Presentation:** 3
**Contribution:** 2
**Rating:** 6
**Confidence:** 3

**Summary:**

This paper introduces a difficulty-aware coding problem generation framework, as well as the corresponding dataset (100K examples). In particular, a teacher / problem generator is trained to produce hard coding problems via (i) a difficulty-aware concept graph and (ii) rejection FT that keeps the hardest-of-K candidates according to a solver-disagreement proxy. Training a student model on the resulting data yields some improvement on hard and average performance.

**Strengths:**

1) The paper proposes training a specialized teacher / problem generator model, rather than prompting a fixed model, to create synthetic code data.
2) The paper includes interesting ablations that anticipate and answer likely reader questions (e.g., Table 3).

**Weaknesses:**

1) The primary novelty of the framework arises from the fine-tuned teacher. Yet, Table 5 shows that the trained teacher performs roughly the same as a fixed teacher (gpt-4o), so the added value (and claimed flexibility) is unclear. Without gains from training a teacher model, the rest of the pipeline/framework  (concept extraction, synthetic data generation, and filtering), largely mirrors prior works.
2) The magnitude of improvement seems small / possibly noisy. In Table 4, RL improves average performance marginally mostly via hard (easy + medium dip in performance when compared to TACO-RL).
3) The paper is missing references/comparisons to other synthetic data generation pipelines that make use of concept extraction (e.g., see the following line of work: https://arxiv.org/abs/2405.12205, https://arxiv.org/abs/2407.21009, https://arxiv.org/abs/2408.14774)

**Questions:**

1) In Table 5, what is the baseline performance for the Qwen model? (In other words, the first row for the gpt-4o as problem generator, but with qwen as the generator.)
2) Teacher and student models are often from the same model family (and even architecture). Could the authors comment on cross-family performance, and how gains reflect the benefits of the framework over the in-family inductive bias?
3) The framework uses an LLM to extract each concept c. How sensitive is the framework to the use of different LLMs for concept extraction?
4) How does the difficulty metric ($\delta$) correlate with human rated difficulty (even on a small sample of the framework data/pipeline)? TACO (the seed dataset) also has human-annotated difficulty ratings.

---

> ### Author Response · Authors · 2025-11-22
>
> Dear reviwer CXvL,
>
> We thank you for your reply, regarding the weakness and questions you mentioned, we have below response.
>
> Weakness 1
> >  The primary novelty of the framework arises from the fine-tuned teacher. Yet, Table 5 shows that the trained teacher performs roughly the same as a fixed teacher (gpt-4o), so the added value (and claimed flexibility) is unclear.
>
> We appreciate the reviewer’s scrutiny of Table 5. While the fine-tuned teacher performs similarly (slightly better) to GPT-4o, we respectfully argue that the primary value of our method lies in efficiency and being model-agnostic, rather than just absolute performance ceilings.
> It is crucial to highlight that our framework achieves better-than GPT-4o-level performance using Qwen2.5-14B-Instruct, a model orders of magnitude smaller than GPT-4o. The fact that our reject-finetuning method allows a 14B open-weights model to slightly outperform a massive proprietary model demonstrates significant value. Furthermore, the performance gain over the original Qwen2.5-14B-Instruct is substantial, proving the method’s effectiveness in elevating base model capabilities.
> Finally, regarding flexibility: our framework is model-agnostic. While computational constraints limited our experiments to the 14B scale, the method is theoretically applicable to stronger foundation models (including GPT-4o itself) to yield further gains.
>
> Weakness 2:
> > The magnitude of improvement seems small / possibly noisy.
>
> We acknowledge that the average improvement in Table 4 appears marginal; however, we observe a distinct performance gain on the 'Hard' subset, which is our core objective, and validates our method's capability to synthesize challenging problems. Furthermore, the RL experiments were conducted on a smaller scale as a proof of concept. The primary validation of our approach lies in the larger-scale distillation experiments. As demonstrated in Table 3 and our new results in Table A, our method consistently outperforms the baselines when scaled up, confirming its robustness.
>
> **Table A, Comparison with TACO using Qwen-235B-A22B as teacher model**
>
> | Student Model | TACO | Ours |
> | :--- | :--- | :--- |
> | Qwen3-8B-Base | 0.4608 | **0.4998** |
>
> **Part of Table 3 in revised paper, 'Our Impl.' means using same strong teacher model (Qwen-235B-A22B) with ours**
>
> | Method | LiveCodeBenchV5 | USACO |
> | :--- | :--- | :--- |
> | OCR-8B (112K, Our Impl.) | 56.5 |  41.3 |
> | QueST-100K-8B (100K, Ours) | 59.4 | 43.5 |
> | QueST-8B (212K, Ours) | **65.2** | **49.9** |
>
> > Weakness 3: The paper is missing references/comparisons to other synthetic data generation pipelines that make use of concept extraction:
>
> We appreciate that you mentioned we have missed several relevant works about concept extraction, we added them in our related work section in our revised version. But we have to emphasize that our focus is not concept extraction, but training difficult aware generators for problem synthesis, and this is the major differences with these previous literatures.

---

> ### Author Response · Authors · 2025-11-23
>
> > Q1. What is the baseline performance for the Qwen model?
>
> We have added the result in Table 5. Results show our method improves substantially over this baseline.
>
> **Part of Table 5 in revised paper,**
>
> | Method | LiveCodeBenchV5|
> | :--- | :--- |
> | Baseline | 32.8  |
> | Baseline w/ difficulty-aware graph | 36.1  |
> | Baseline w/ difficulty-aware graph w/ RFT (QueST) | **37.9** |
>
> > Q2. Could the authors comment on cross-family performance, and how gains reflect the benefits of the framework over the in-family inductive bias?
>
> We appreciate this valuable suggestion. We agree that testing cross-family performance is an interesting direction to isolate the framework's benefits from in-family inductive bias. However, our current experiments follow the standard setting used by leading open-source models (e.g., the Qwen series, where the 8B model is distilled from the larger model within the same family). Due to computational constraints, we prioritized this prevalent setting, but we will consider cross-architecture experiments in future work.
>
>  > Q3. How sensitive is the framework to the use of different LLMs for concept extraction
>
> This is an interesting question that also pertains to previous work using concept extraction (such as MathScale[1], PromptCoT[2], and Pseudo Feedback[3]), Pseudo feedback[3] already used GPT-4o for concept extraction, so we follow their setting. And we consider a systematic analysis of concept extraction outside the scope of our contribution.
>
>
> >Q4. How does the difficulty metric $\delta$ correlate with human rated difficulty
>
> We compare Pearson and Spearman correlation in Table B across using different models to compute the difficulty score. The results demonstrate a clear positive correlation, confirming that our metric aligns well with human-perceived difficulty.

---

### Official Review · Reviewer_vYVu · 2025-11-02

**Soundness:** 3
**Presentation:** 3
**Contribution:** 2
**Rating:** 6
**Confidence:** 3

**Summary:**

The paper proposes QueST, a framework to train problem-generating LLMs that produce hard competitive-programming questions at scale. Two core ideas:

1) a difficulty-aware concept graph that biases prompt construction toward concept pairs associated with higher human difficulty labels in the seed set (TACO), implemented by blending co-occurrence with average difficulty in the edge weights (Eq. 10; α=0.2) ; and

2) difficulty-aware rejection fine-tuning (RFT) that, for each prompt, samples multiple questions and keeps the hardest one under a difficulty proxy δ defined as one minus the average majority-vote rate across M candidate solutions over T test inputs (Eq. 6) .

QueST uses GPT-4o to generate test inputs and candidate solutions while computing δ, filters low-quality cases, and trains a specialized generator (Qwen2.5-14B-Instruct) with RFT. The resulting synthetic questions (20k–100k) are paired with long-CoT solutions from Qwen3-8B to SFT smaller students (Qwen2.5-Coder-7B-Instruct, Qwen3-8B-Base), and also support RLVR training (GRPO). On LiveCodeBench-V5 and USACO, QueST variants outperform baselines of similar size, with the 8B student showing the strongest average gains (Table 3)

**Strengths:**

1) Clear, modular pipeline with explicit math and algorithms. The $\delta$ metric and RFT selection are precisely defined (Eqs. 5–9), with a practical filtering step for invalid executions

2) The edge-weighting scheme combining co-occurrence and average difficulty is easy to implement and justified by seed annotations in TACO

3) Training on QueST-generated data with Qwen3-8B as teacher matches or exceeds prior SFT datasets that used larger/stronger teachers, especially on harder USACO levels (Table 3)

4) Contamination check reported. 50-gram Jaccard similarity is 0 across used datasets/benchmarks (per their method), which is appreciated

**Weaknesses:**

1) Difficulty proxy $\delta$ may conflate “hardness” with generator/judge idiosyncrasies; causal link not isolated.

2) The paper does not evaluate $\delta$ stability across different judge models.

3) The paper itself cites difficulty-aware/rejection methods and concept-graph generation (e.g., MathScale; DART-math; "weakness-driven" synthesis) in related work; QueST’s novelty is the combination plus code-specific engineering, not the first instance of difficulty-aware synthetic problem generation (Sec. 4)

4) Table 3 shows the 7B model trained on QueST-100K is slightly below OCR-Qwen-7B-Instruct on LiveCodeBench Avg (43.3 vs 51.3), though it performs relatively better on USACO-Hard. The narrative should be more calibrated: QueST helps, but it's not uniformly superior to the best existing 7B post-training recipe.

**Questions:**

1) Does $\delta$  correlate with human difficulty? Please report Spearman/Pearson between $\delta$  and TACO’s human-labeled difficulty on held-out problems, and across judge models

2) How do results change with different M (number of solutions), T (test inputs), temperature, and different solvers for computing $\delta$?

3) Can you provide a small human or symbolic audit to estimate the precision of high-$\delta$ items (i.e., % well-posed with correct reference outputs)? Even 100 sampled problems would help.

---

> ### Author Response · Authors · 2025-11-22
>
> Dear reviewer vYVu,
>
> We appreciate your valuable reply, regarding the weakness and questions you mentioned, we reply as below:
>
> > Weakness 1 Difficulty proxy $\delta$
>  may conflate “hardness” with generator/judge idiosyncrasies; causal link not isolated.:
>
>  While we acknowledge that isolating a strict causal link is challenging, we offer two lines of evidence to support the validity of our difficulty proxy:
>
> Previous literature discussion: As discussed in Section 2.2, we align with established findings in [1] (SC-CoT) which demonstrate that model consistency is a indicator of problem difficulty and highly correlated with accuracy.
>
> Empirical results: To directly address the concern that our proxy captures generator-specific quirks, we conducted additional experiments using varous generator models. As illustrated in Figure 3 of the revised pdf and Table (see below), we observe that the positive correlation between our hardness proxy and consistency remains statistically significant across all tested models. This cross-model consistency suggests that our metric captures the intrinsic difficulty of the problem rather than the idiosyncrasies of a single generator or judge.
>
> **Table B**: Correlations with human-rated difficulty (From TACO, we transform difficulty levels to 1-5 scores, corresponding to easy, medium, medium hard, hard, and very hard, respectively.)
>
>
> | Correlations/Models | GPT-4o | Qwen3-8B | Qwen3-30B-A3B-Instruct-2507 |
> | :--- | :--- | :--- | :--- |
> | Spearman | 0.38, p-value=1.9e-8 | 0.54, p-value=9e-15 | 0.51, p-value=9e-15 |
> | Pearson | 0.39, p-value=1.5e-8 | 0.51, p-value=8e-17 | 0.49, p-value=6e-14 |
>
> > Weakness 2 The paper does not evaluate
>  stability across different judge models:
>
> If we interpret "judge model" as the model employed to calculate output consistency (via majority voting), we have conducted additional ablation studies to evaluate the stability of $\delta$.
> We repeated our measurements using different backbone models for consistency estimation on a representative subset of the data. As illustrated in the newly added Table B (see above) and Figure 3 in the paper, correlations are relatively stable between Qwen3-8B and Qwen3-30B-A3B-Instruct-2507. We observe a lower, but still positive correlation for GPT-4o. This confirms that it serves as a stable and model-agnostic indicator. Since Qwen3-8B is a computationally intensive long CoT model, we used GPT-4o APIs for large-scale output generation.
>
> > Weakness 3 The paper itself cites difficulty-aware/rejection methods and concept-graph generation ... QueST’s novelty is the combination plus code-specific engineering, not the first instance of difficulty-aware synthetic problem generation:
>
> We respectfully disagree that QueST is merely a 'combination plus code-specific engineering.' While we build upon related concepts, our method differs fundamentally from the cited works in the following ways:
>
> Vs. MathScale: MathScale utilizes standard graph construction prompts to generate problems. In contrast, QueST introduces a novel difficulty-aware graph sampling algorithm, allowing for precise control over problem complexity during generation.
>
> Vs. DART-Math: DART-Math is a data selection strategy that rebalances training on existing difficult problems; it synthesizes responses, but not questions, in contrast to our work, which focuses on questions synthesis.
>
> Vs. 'Weakness-driven' synthesis: This approach does not involve training generators to create synthetic problems.
> Therefore, QueST is the first framework to propose difficulty-aware graph sampling and the first to train generators using difficulty-aware rejection fine-tuning. These contributions go beyond simple engineering and represent a distinct methodological advancement.
>
> > Weakness 4: QueST helps, but it's not uniformly superior to the best existing 7B post-training recipe.
>
> We appreciate the reviewer’s careful examination of Table 3. We would like to clarify that the results in this table do not represent a strictly fair comparison, as QueST-100K operates under a significant resource disadvantage compared to OCR-Qwen-7B. Specifically, our model was trained on far less data (100K vs. 700K) and utilized a weaker teacher model (Qwen3-8B vs. DeepSeek-R1).
>
> Our original intention with Table 3 was to demonstrate data efficiency, showing that QueST can achieve competitive performance despite these constraints. To address the reviewer’s concern regarding a fair comparison, we have added new results using a scaled-up setting (using Qwen-235B-A22B as teacher) and conducted fair comparison with using similar size OCR data and same strong teacher models in revised Table 3 in paper. These results demonstrate that when trained under comparable conditions, QueST outperforms OCR with same teacher model, and achieve state-of-the-art performance in LCB among 8B sized open-weight models.
>
> [1] Self-Consistency Improves Chain of Thought Reasoning in Language Models. Wang et al.

---

> ### Author Response · Authors · 2025-11-22
>
> > Q1:Does $\delta$
>  correlate with human difficulty? Please report Spearman/Pearson between
>  and TACO’s human-labeled difficulty on held-out problems, and across judge models
>
> We report the requested Spearman and Pearson correlations between our estimated difficulty and TACO’s human-labeled difficulty, as well as the consistency across different judge models, in Table B. The results demonstrate a strong alignment between our method and human judgment, validating the reliability of our difficulty estimation.
>
> > Q2 How do results change with different M (number of solutions), T (test inputs), temperature, and different solvers for computing $\delta$?:
>
> We assessed the stability of our estimation by computing correlations across varying M and T values and different solvers. The results show consistent performance, confirming that our difficulty estimation remains robust even with limited samples or different backbone models. And shows clear trend that when number of test cases and solutions becomes larger, the correlation with human labelled difficulty also become larger.
>
> Added results for different **M and T** when compute estimated difficulty:
>
> **Varying solutions number (M)**
>
> | Different solutions number (10 testcases) | 4 | 6 | 8 |
> | :--- | :--- | :--- | :--- |
> | Pearson | 0.325 | 0.366 | 0.389 |
> | Spearman | 0.327 | 0.361 | 0.381 |
>
>
> **Varying testcases number (T)**
>
> | Different testcases (8 solutions) | 6 | 8 | 10 |
> | :--- | :--- | :--- | :--- |
> | Pearson | 0.371 | 0.381 | 0.389 |
> | Spearman | 0.367 | 0.378 | 0.381 |
>
> > Q3 Can you provide a small human or symbolic audit to estimate the precision of high-$\delta$
>  items (i.e., % well-posed with correct reference outputs)? Even 100 sampled problems would help.:
>
> We appreciate the suggestion to estimate the precision of high-theta items. However, we note that a manual audit of 100 items is practically infeasible due to the complexity of these problems. Since 'high-theta' items correspond to difficult competitive programming tasks, verifying a single problem (including code logic and corner cases) typically requires 30 to 60 minutes of expert time. A 100-item audit would therefore necessitate 50–100 hours of expert labor, which is beyond the scope of the rebuttal period.

---

### Author Response · Authors · 2025-11-23
**General Response, Clarifications, and New Results For Common Questions in All Reviewers**

We thank the reviewers for their valuable comments. As several major questions overlap, particularly regarding our new results, we provide this general response to address them collectively.

**1. Regarding the Difficulty Estimation Method**

Reviewers vYVu, CXvL, and jBTM all raised questions regarding our difficulty estimation method. specifically concerning the stability of estimated difficulty (vYVu), its correlation with human ratings (vYVu, CXvL), and the lack of theoretical justification beyond self-consistency (jBTM).
To address this, we performed additional analysis using outputs from GPT-4o, Qwen3-30B-A3B-Instruct-2507, and Qwen3-8B. We calculated the difficulty using our method and correlations with human ratings in the TACO dataset (Table B). We observed that the positive correlation between our hardness proxy and consistency remains statistically significant across all models. These correlations are stable; for GPT-4o, the correlation is lower but remains positive. Since Qwen3-8B is a computationally intensive long CoT model, we used GPT-4o APIs for large-scale output generation. We also draw figure 3 in paper (line 486-499) to further show same trend between models.

**Table B: Correlations with human-rated difficulty (From TACO, we transform difficulty levels to 1-5 scores, corresponding to easy, medium, medium hard, hard, and very hard, respectively.)**


| Correlations/Models | GPT-4o | Qwen3-8B | Qwen3-30B-A3B-Instruct-2507 |
| :--- | :--- | :--- | :--- |
| Spearman | 0.38, p-value=1.9e-8 | 0.54, p-value=9e-15 | 0.51, p-value=9e-15 |
| Pearson | 0.39, p-value=1.5e-8 | 0.51, p-value=8e-17 | 0.49, p-value=6e-14 |


**2. Regarding the Performance of Our Method**

Second, all reviewers raised questions regarding the performance comparison of our method against prior approaches. Specifically, Reviewer vYVu noted that our method "is not uniformly superior to the best existing 7B post-training recipe," while Reviewer CXvL remarked that the "magnitude of improvement seems small / possibly noisy." Reviewer jBTM stated that the "synthetic data do not show clear superiority," and reviewer HaRZ added that "the authors cannot explain the reason for the low accuracy in the synthetic problems," and asking how QueST compares to TACO when data volume is controlled.

To address these concerns, we conducted further experiments to validate the superiority of our method. We applied our method using a stronger teacher model (Qwen3-235B-A22B). To ensure a fair comparison, we re-implemented the OCR method using Qwen3-8B-Base as the student model and generated 4 responses per problem using Qwen3-235B-A22B as the teacher (yielding a total of 112K examples).
Even under these conditions, QueST-100K-8B outperforms OCR-8B across both benchmarks. This suggests that a large volume of diverse, challenging synthetic problems provides greater benefit than simply repeating existing human-written ones. Furthermore, by combining the 112K organic examples with our 100K generated examples, we achieve even stronger results (QueST-8B), attaining performance on par with the much larger DeepSeek-R1-671B.

**Table 3 Full Results Comparison With Previous Baselines and Stronger Settings in Our implementations.** (line 324-347)
| Model | LCB-V5 Easy | LCB-V5 Medium | LCB-V5 Hard | LCB-V5 Avg. | USACO Easy | USACO Medium | USACO Hard | USACO Avg. |
| :--- | :---: | :---: | :---: | :---: | :---: | :---: | :---: | :---: |
| DeepSeek-R1-671B (Guo et al., 2025) | 98.5 | 79.8 | 37.4 | 65.6 | 72.5 | 54.6 | 34.3 | 56.2 |
| Qwen3-8B (Yang et al., 2025) | 94.0 | 74.1 | 28.9 | 58.7 | 58.5 | 42.8 | 22.3 | 43.5 |
| R1-0528-Qwen3-8B (Guo et al., 2025) | 94.4 | 73.5 | 27.7 | 58.1 | 57.0 | 33.6 | 17.2 | 38.5 |
| OpenThinker-7B (114K) (Guha et al., 2025) | 80.6 | 16.9 | 1.6 | 25.5 | 11.0 | 2.1 | 0.0 | 5.0 |
| R1-Distill-Qwen-7B (800K) (Guo et al., 2025) | 86.6 | 43.8 | 7.0 | 38.0 | 22.9 | 9.7 | 3.8 | 13.4 |
| OlympicCoder-7B (100K) (Face, 2025) | 82.1 | 49.4 | 12.2 | 40.9 | 31.4 | 12.5 | 1.3 | 17.0 |
| OCR-Qwen-7B-Instruct (700K) (Ahmad et al., 2025) | 95.4 | 64.0 | 18.0 | 51.3 | 41.5 | 26.0 | 7.5 | 27.2 |
| OCR-8B (112K, Our Impl.) | 96.0 | 70.2 | 26.2 | 56.5 | 54.5 | 40.5 | 22.9 | 41.3 |
| QueST-100K-8B (100K, Ours) | 97.1 | 74.8 | 28.4 | 59.4 | 55.9 | 44.0 | 24.7 | 43.5 |
| **QueST-8B (212K, Ours)** | **97.6** | **81.0** | **36.6** | **65.2** | **65.5** | **48.6** | **28.7** | **49.9** |

We also present a controlled comparison between QueST and TACO using identical data volumes. The results demonstrate that QueST outperforms TACO under this setting. This empirical evidence supports the conclusion that the performance improvements are driven by the superior quality and complexity of the QueST problems, rather than data scale.

**Table A, Comparison with TACO using Qwen-235B-A22B as teacher model. Pass@1 in LiveCodeBenchV5**

| Student Model | TACO | Ours |
| :--- | :--- | :--- |
| Qwen3-8B-Base | 0.4608 | **0.4998** |

---

### Author Response · Authors · 2025-12-03
**TL;DR, Summary of Responses.**

Dear AC,

We thank you for your time and effort, especially during the special scenario this year.
As three out of four reviewers have not yet replied to our detailed responses, we would like to provide a final summary confirming that we have addressed the most critical limitations and questions raised. We hope this serves as a helpful reference for your evaluation.

We have primarily incorporated additional results and analyses covering two key aspects, which were common concerns among the reviewers:

- **1. Difficulty Estimation Method (Raised by vYVu [W1, W2], CXvL [Q4], jBTM [W2])**

   Concerns were raised regarding whether our difficulty estimation method correlates with human-labeled difficulty and the stability. We added **Figure 3** to the revised paper, demonstrating a similar trend across different models in terms of Avg Majority Voting Rate vs. Human Rating Level. Additionally, we added **Table B** (see below comments) to statistically prove positive correlations with human-labeled difficulty.

- **2. Performance of Our Method (Raised by vYVu [W4], CXvL [W2], jBTM [W3], HaRZ [W2, W3])**

    We added **Table A** (see below comments) to provide a fair comparison with using TACO's problems. We also included scaling-up experiments in the revised **Table 3** to ensure a fair comparison with OCR and other previous methods. These results demonstrate that our 8B model achieves scores comparable to DeepSeek-R1-671B.

**Summary of our specific discussions with each reviewer**:

**Reviewer vYVu (Score: 6, No follow-up)**

- **W1 & Q1**: We added Table B and Figure 3 to the revised paper, demonstrating a consistent positive correlation across three different models.

- **W2**: We added Table B and Figure 3 to show that stability is maintained across various models.

- **W3**: We emphasized the distinctions between our method and previous approaches, clarifying misunderstandings regarding the methodology.

- **W4**: We added new results in Table 3, showing that we achieve state-of-the-art performance on LCB among similar-sized models.

- **Q2**: We included robustness experiments across various M and T, confirming that our difficulty estimation method is robust.

**Reviewer CXvL (Score: 6, No follow-up)**

- **W1**: We clarified that the primary value of our method lies in its efficiency and model-agnostic nature. We utilized Qwen-2.5-14B, which outperforms GPT-4o in this context.

- **W2**: We added the results of Table A and the revised Table 3 to the paper.

- **W3**: We added the missing references to our revised paper.

- **Q1**: We added baseline results in the revised Table 5.

- **Q2**: We clarified that our current experiments follow the standard settings used by leading open-source models.

- **Q3**: We explained that a systematic analysis of concept extraction is outside the scope of our contribution. Additionally, we noted that additional human labeling would require 50–100 hours, which is beyond the scope of the rebuttal period.

- **Q4**: We added the results of Table B to directly address the correlation question.

**Reviewer jBTM (Score: 4, No follow-up)**

- **W1**: We clarified that MathScale serves only as a scaffold within our pipeline. Our primary contribution is a novel method for difficulty-aware problem synthesis.

- **W2**: We addressed this concern from three perspectives: theoretical grounding in prior literature (SC-CoT), empirical alignment with human judgment (Table B and Figure 3), and downstream task effectiveness (Table A and Table 3).

- **W3**: We addressed the performance concern by adding new results in Table 3.

- **W4**: We clarified that utilizing seed data to bootstrap the generation of complex queries is a standard and practical setting.

- **Q1**: We explained that our primary focus is difficulty estimation.

**Reviewer HaRZ (Score: 2, Follow-up addressed, awaiting response)**
- **W1 & Q1**: We clarified that our model choices were not arbitrary and provided our rationale.

- **W2**: We sought clarification regarding the mention of "low accuracy in the synthetic problems," as the reviewer has not yet replied. However, we added results in Table A and Table 3 to demonstrate that our performance is superior to previous methods.

- **W3 & Q2**: We added the performance of generator models on coding tasks, showing that their performance is significantly lower than our trained model.

- **Q3**: We noted that the model in question is the output of a specific synthetic SFT data creation method rather than a backbone model.

- **Q4**: We added Table B to address the difficulty estimation question.

- **Q5**: We clarified our performance regarding the RL results and Table 3.

- **Follow-up question**: We demonstrated that our QueST-7B substantially outperforms Qwen2.5-14B-Instruct. Therefore, it is reasonable to expect that using a stronger student model (e.g., Qwen2.5-14B-Instruct) would yield results that exceed the current QueST-7B score.

Best regards,

The Authors

---

### Meta-Review · Area_Chair_d2Hu · 2026-01-13

**Summary:**

This submission proposes QueST, a difficulty-aware pipeline for generating challenging competitive-programming problems and using the resulting synthetic data to post-train downstream coding models. Reviewers generally found the pipeline clearly described and the topic timely, but the decision-critical concerns were that the core difficulty proxy and selection mechanism remain only partially validated as measuring “intrinsic” difficulty, the empirical improvements were viewed as not consistently compelling relative to strong existing baselines and are sensitive to experimental choices, and the paper’s novelty is best interpreted as an integration/engineering of known ingredients rather than a clearly distinct methodological advance. These issues, together with remaining questions about data quality and the dependence on substantial external model infrastructure, outweigh the strengths and keep the overall evaluation below the acceptance threshold.

**Reviewer Concerns:**

The rebuttal adds correlation analyses with human-labeled difficulty and provides additional controlled comparisons, which help clarify the authors’ position; however, across the reviews the central skepticism is not fully resolved in a way that would change the overall assessment. In particular, the evidence for the difficulty metric remains correlational and does not convincingly disentangle “hardness” from generator/judge idiosyncrasies or other confounders, and the request for a concrete quality check of high-difficulty items is still unanswered in practice, leaving ambiguity about how much of the retained set reflects genuine complexity versus noise or subtle errors. On performance, while the authors provide additional results and argue fairness via stronger-teacher re-implementations and data-volume controls, reviewers’ original concern—that gains are not uniformly superior and can look marginal or baseline-dependent—remains salient, especially given the number of moving parts. Finally, the novelty concern persists: the method appears to build on prior concept-graph and difficulty-aware/rejection-style synthesis ideas, and the rebuttal does not fully establish a clear conceptual advance beyond a task-specific composition and scaling of existing components; as a result, the contribution is perceived as incremental and insufficiently compelling for acceptance at ICLR.

**Reviewer Scores:**

Given the remaining gaps and the reviewers’ stated positions, I anticipate all reviewers would keep their scores unchanged if they had participated fully in further discussion: vYVu stays at 6, as the rebuttal improves clarity but does not eliminate their core concerns about isolating the difficulty proxy and calibrating claims versus best post-training recipes; CXvL stays at 6, as additional baselines/references help but do not remove the impression that improvements are small/noisy and that the fine-tuned teacher’s added value is not decisively demonstrated; jBTM stays at 4, as the rebuttal does not sufficiently change their view that the approach is a heuristic extension of prior pipelines with limited demonstrated superiority; and HaRZ stays at 2, given continued concern about data quality/self-improvement and missing clarity on comparisons such as “direct distillation,” alongside broader doubts about the pipeline’s assumptions and evidence.

---

### Decision · Program_Chairs · 2026-01-26

Reject